# Monogenic Parkinson’s Disease: Genotype, Phenotype, Pathophysiology, and Genetic Testing

**DOI:** 10.3390/genes13030471

**Published:** 2022-03-07

**Authors:** Fangzhi Jia, Avi Fellner, Kishore Raj Kumar

**Affiliations:** 1Department of Neurology, Concord Repatriation General Hospital, Concord, NSW 2139, Australia; fangzhi.jia@hotmail.com; 2Sydney Medical School, University of Sydney, Camperdown, NSW 2050, Australia; 3Garvan Institute of Medical Research, Darlinghurst, NSW 2010, Australia; avi.fellner@gmail.com; 4Raphael Recanati Genetics Institute, Rabin Medical Center, Beilinson Hospital, Petah Tikva 4941492, Israel; 5Department of Neurology, Rabin Medical Center, Beilinson Hospital, Petah Tikva 4941492, Israel; 6Molecular Medicine Laboratory, Concord Repatriation General Hospital, Concord, NSW 2139, Australia

**Keywords:** monogenic, Parkinson’s disease, genomics, genetic testing, deep brain stimulation

## Abstract

Parkinson’s disease may be caused by a single pathogenic variant (monogenic) in 5–10% of cases, but investigation of these disorders provides valuable pathophysiological insights. In this review, we discuss each genetic form with a focus on genotype, phenotype, pathophysiology, and the geographic and ethnic distribution. Well-established Parkinson’s disease genes include autosomal dominant forms (*SNCA*, *LRRK2*, and *VPS35*) and autosomal recessive forms (*PRKN*, *PINK1* and *DJ1*). Furthermore, mutations in the *GBA* gene are a key risk factor for Parkinson’s disease, and there have been major developments for X-linked dystonia parkinsonism. Moreover, atypical or complex parkinsonism may be due to mutations in genes such as *ATP13A2*, *DCTN1*, *DNAJC6*, *FBXO7*, *PLA2G6*, and *SYNJ1*. Furthermore, numerous genes have recently been implicated in Parkinson’s disease, such as *CHCHD2*, *LRP10*, *TMEM230*, *UQCRC1*, and *VPS13C*. Additionally, we discuss the role of heterozygous mutations in autosomal recessive genes, the effect of having mutations in two Parkinson’s disease genes, the outcome of deep brain stimulation, and the role of genetic testing. We highlight that monogenic Parkinson’s disease is influenced by ethnicity and geographical differences, reinforcing the need for global efforts to pool large numbers of patients and identify novel candidate genes.

## 1. Introduction

Parkinson’s disease (PD) is a common neurodegenerative disorder in which we have an incomplete understanding of the molecular and cellular disease basis and no currently available disease-modifying therapy. A key strategy to understanding the pathogenesis of PD is to investigate the underlying genetic basis. Approximately 5–10% of PD can be attributed to monogenic forms. Other causes are related to a combination of complex genetic susceptibility and environmental factors. For the monogenic forms, there are several well-established genes, with autosomal dominant (*SNCA*, *LRRK2*, and *VPS35*) and autosomal recessive (*PRKN*, *PINK1*, *DJ1*) modes of inheritance. Additionally, there is X-linked inheritance (X-linked dystonia-parkinsonism) and atypical or complex parkinsonian phenotypes due to mutations in the *ATP13A2*, *DCTN1*, *DNAJC6*, *FBXO7*, *PLA2G6*, and *SYNJ1* genes. Moreover, there are numerous recently reported genes, including *CHCHD2*, *LRP10*, *TMEM230*, *UQCRC1*, and *VPS13C*. In some cases, the same gene can be linked with Mendelian forms of PD as well as increased susceptibility (such as *SNCA* and *LRRK2*). Furthermore, mutations in genes such as glucocerebrosidase (*GBA*) fall between a monogenic cause and a genetic susceptibility factor [1]. Further recent discoveries have focused on the clinicogenetic and pathological findings, which will be discussed. Pathophysiological insights will be discussed briefly (for a more detailed discussion, see elsewhere [2]), and a discussion of novel therapeutic candidates can be found elsewhere [3,4]. Recent hot topics include the understanding of the effect of heterozygous variants in recessive PD genes, outcomes in individuals who co-inherit mutations in both *GBA* and *LRRK2*, the effect of the underlying genetic form on outcomes from deep brain stimulation, and regional and ethnic differences for mutations in PD genes. In this review, we provide a concise summary of the monogenic origin of PD with a focus on these recent developments.

## 2. Autosomal Dominant Forms

### 2.1. SNCA

#### 2.1.1. Genotype-Phenotype

*SNCA* mutations cause autosomal dominant PD and can be due to different mutation types, including missense variants and multiplications (Table 1). So far, there have been eight missense variants identified as causing autosomal dominant PD: p.A30G, p.A30P, p.E46K, p.H50Q, p.G51D, p.A53E, p.A53T, and p.A53V. An MDSGene review identified phenotypic differences between some of these missense variants, with the most common mutation p.A53T having an early age at onset compared to p.A30P and p.E46K [5]. However, these findings are uncertain given that the number of cases for *SNCA* missense variants other than p.A53T is small [5]. Of note, there is evidence that the p.H50Q variant is not enriched in cases versus controls and thus may not have sufficient evidence to be considered pathogenic [6]. The most recently reported mutation, p.A30G, was found in five affected individuals from three Greek families [7]. The phenotype results in a widely ranging age at onset, an initial good response to medication (Table 2), prominent motor fluctuations, and a range of non-motor manifestations such as orthostatic hypotension, REM-behavior sleep disorder, cognitive impairment, and psychiatric manifestations [7].

Duplications and triplications of the *SNCA* gene can also cause PD. *SNCA* duplications cause a phenotype resembling idiopathic PD, whereas *SNCA* triplications cause rapidly progressive PD with earlier onset and extensive Lewy Body pathology. A recent study highlighted the correlation between *SNCA* dosage and age at onset, with copy number 3 (heterozygous *SNCA* duplication) associated with a mean age at onset of 46.9 ± 10.5 years versus copy number 4 (homozygous *SNCA* duplication, or *SNCA* triplication) associated with a mean age at onset of 34.5 ± 7.4 [8].

Overall, duplications are more common than missense mutations or triplications [5]. The different mutation types can be stratified according to the age at onset, with early, intermediate, and late onset for triplications, missense mutations, and duplications, respectively [5].
genes-13-00471-t001_Table 1Table 1Genotype-phenotype summary for monogenic forms of Parkinson’s disease.GeneMode ofInheritanceFrequencyEthnic PopulationDistributionTypes ofMutationsClinicalPhenotypeResponse to PDMedicationResponse to DBSPathologicalFindings*SNCA*ADRare, with a frequency from 0.045% to 1.1% in recent studies [9]Majority European, then Asians and Hispanics [5]Missense, duplications, and triplicationsRange of age at onset, prominent motor fluctuations, range of complications including cognitive impairment and psychiatric manifestationsInitial good responseFew examples, appears to have a good response for duplications, poor response for missense mutationsα-synuclein-positive and LB pathology [10]*LRRK2*AD1% of PD but higher in North African Berber Arab and Ashkenazi Jewish populationsThe p.G2019S mutation found in Europeans with high prevalence in North African Berbers and Ashkenazi Jews7 missense variants describedResembles idiopathic PDVast majority show a good response to levodopa [11]DBS is effective [12]Most patients with the p.G2019S mutation show LB pathology, whereas this finding is rare for other mutations*VPS35*ADRare (overall prevalence of 0.115%)European, Asian, Ashkenazi Jewish [5]1 missense mutation described, p.D620NResembles idiopathic PDGood response [11]Small numbers reported, at least 2 had a good outcome [11]Not available*PRKN*ARMost common cause of EOPD, 12.5% of recessive PD [13]Majority Asian, followed by Caucasians and Hispanics [14]Missense mutations, frameshift mutations, structural variantsEOPD, lower limb dystonia, absence of cognitive impairmentGood response to levodopa therapy, frequent motor fluctuations, and dyskinesiasGood outcome in all patients [11]Substantia nigra pars compacta loss with the notable absence of LB pathology*PINK1*ARSecond most common cause of EOPD, 1.9% of recessive PD [13]European, Asian, may be frequent in Arab Berber and Polynesian populations [9,14,15]Missense mutations, nonsense mutations, structural variantsEOPD, typical PD, dyskinesias, dystonia, and motor fluctuations can occurVast majority show a good outcome [11]Good or moderate [11]LB pathology may or may not be present in the handful of autopsy cases reported*PARK7* (*DJ1*)AR0.16% of recessive PD [13]Most patients are from Italy, Iran, and Turkey [14]Missense, splice site, frameshift, and structural variantsEOPD50% show a good response, others moderate or minimal [11]No reports identified [11]LB pathology [16]*TAF1*X-linked0.34 per 100,000 in the Philippines, Island of Panay 5.24 per 100,000Philippines, high prevalence on the Island of PanayInsertion of a SINE-VNTR-Alu type retrotransposon in intron 32 of the *TAF1* geneParkinsonism, dystoniaMay be responsive to levodopa, particularly for those with pure parkinsonism [17]DBS results in an improvement in dystonia and to a lesser extent parkinsonism [18]Accumulation of lipofuscin in the neurons and glia, but absence of LB pathology*ATP13A2*ARRareSpread across the globe [19]Frameshift, missense, and splice site mutations [19]KRS, clinical triad of spasticity, dementia, and supranuclear gaze palsy [20], facial-faucial-finger mini-myoclonus [21], other phenotypes include HSPVariable response to levodopa [19]May respond well, variable [22]Accumulation of lipofuscin, absence of LB pathology [23]*DCTN1*ADRareSpread across the globe10 different heterozygous missense mutations [19]Perry syndrome—rapidly progressive parkinsonism, depression and mood changes, weight loss, and progressive respiratory changesMay be levodopa-responsiveNo reports identifiedSelective loss of putative respiratory neurons in the ventrolateral medulla and in the raphe nucleus, no or few LBs, TDP43-positive inclusions [24,25]*DNAJC6*ARRareMainly found in Middle Eastern populations, although families of European origin have also been found to harbor *DNAJC6* mutations5 different homozygous mutations, largest family carries a nonsense mutation [19]Juvenile PD with complicating features, EOPDPoorGood outcome [26]No reports identified*FBXO7*ARRareReported in the Iranian, Turkish, Italian, Dutch, Pakistani, and Chinese populationsBiallelic missense, splice site, and nonsense mutationsJuvenile PD, EOPD, parknsonian-pyramidal syndrome, can overlap with NBIA [27]VariableNo reports identifiedNo reports identified*PLA2G6*ARRareVarious ethnic groups, including Indian, Pakistani, European, Japanese, Chinese, and Korean populations54 mutations associated with parkinsonism [28]Adult-onset dystonia-parkinsonism with cognitive and psychiatric symptoms [28], other phenotypes include NBIAVariableMay benefit from DBS [28]Mixed Lewy and Tau pathology [28]*SYNJ1*ARRareReported in Iranian, Italian, German, Algerian, Senegalese, and Chinese populationsMissense, frameshiftParkinsonism in the third decade of life, complicating features such as dystonia, seizures, or cognitive impairmentPoorNo reports identifiedNo reports identified*CHCHD2*ADRareJapanese and Chinese patientsMissense, splice siteTypical PDGoodNo reports identifiedA brain autopsy revealed widespread α-synuclein pathology with Lewy bodies present in the brainstem, neocortex, and limbic regions [29]*LRP10*ADRareItaly, TaiwanLoss of function and missense variantsLate onset PD, PD dementia, dementia with Lewy Bodies [30,31,32]Good [32]Excellent response for a patient with a *LRP10* and *GBA* variant in trans [33]Severe LB pathology*TMEM230*ADRareIdentified in a Canadian Mennonite familyMissense variantTypical PDResponds to levodopa in most casesNo report identifiedTypical LB pathology [34]*UQCRC1*ADRareTaiwan, may not be in European populationsMissense variantsParkinsonism with polyneuropathyGoodNo report identifiedNo report identified*VPS13C*ARRareTurkish, FrenchTruncating mutationsEOPD, rapid progression, complicating features including dysphagia, cognitive impairment, hyperreflexia [19,35]Initial good responsePoorResembles diffuse LB disease [35]AD: autosomal dominant, AR: autosomal recessive, DBS: deep brain stimulation, EOPD: early-onset Parkinson’s disease, HSP: hereditary spastic paraplegia, KRS: Kufor Rakeb syndrome, LB: Lewy body, NBIA: neurodegeneration with brain iron accumulation, PD: Parkinson’s disease, XDP: X-linked dystonia parkinsonism.


#### 2.1.2. Pathophysiology

The discovery of dominant mutations in *SNCA* as a cause of PD is consistent with the critical role the α-synuclein protein plays in PD pathogenesis. The molecular effects may vary according to the type of *SNCA* mutation [36]. The p.A30P, p.A53T, and p.E46K mutations all affect the N-terminal domain of the α-synuclein protein [36]. The p.A30P and p.A53T mutations stimulate protofibril formation and smaller to larger aggregates [36]. The p.E46K mutation increases the N-terminal positive charge and enhances N-terminal and C-terminal contacts, whereas the opposite is seen for the p.A30P and p.A53T mutations [36]. A recent study showed impaired mitochondrial respiration, energy deficits, vulnerability to rotenone, and altered lipid metabolism in dopaminergic neurons derived from a patient with the p.A30P mutation in *SNCA,* with a comparison to gene-corrected clones, highlighting the numerous effects of these mutations [37].genes-13-00471-t002_Table 2Table 2Levodopa-responsiveness stratified according to Parkinson’s disease monogenic forms.Good Response to LevodopaPoor, Variable, or Uncertain Response to Levodopa*SNCA**TAF1**LRRK2**ATP13A2**VPS35**DCTN1**PRKN**DNAJC6**PINK1**FBXO7**DJ1**PLA2G6**CHCHD2**SYNJ1**LRP10*
*TMEM230*
*UQCRC1*
*VPS13C*


### 2.2. LRRK2

#### 2.2.1. Genotype-Phenotype

At least seven missense variants in *LRRK2* have been described as causing PD (p.N1437H, p.R1441C/G/H, p.Y1699C, p.G2019S, and p.I2020T) [3]. On an individual level, *LRRK2*-PD is clinically indistinguishable from idiopathic PD. However, as a group, it may be considered as having a milder phenotype [38,39]. For example, *LRRK2* mutation carriers are less likely to have non-motor symptoms such as olfactory impairment, cognitive features, and REM-behavior sleep disorder [39]. Furthermore, patients with *LRRK2*-PD may be susceptible to certain cancers [40,41,42]. A very recent study provides evidence that *LRRK2*-PD is associated with a significantly higher risk of stroke [43]. Additionally, recent evidence suggests that regular use of non-steroidal anti-inflammatory drugs may be associated with reduced penetrance of PD in both pathogenic and risk variant carriers [44].

The most common and well-characterized *LRRK2* mutation is the p.G2019S mutation. It has a prevalence of 1% in the PD population with a high prevalence in North African Berber Arab (39%) and Ashkenazi Jewish (approximately 18%) populations [45,46,47]. The penetrance of this mutation is incomplete and variable and influenced by age, environment, and genetic background [48].

Other mutations in *LRRK2* may be relevant to different ethnic and regional populations. For example, the p.R1441C variant has a founder effect in Basque populations and may be higher in Southern Italy and Belgium [38]. The p.G2019S mutation is very rare in Chinese populations, whereas the p.G2385R and p.R1628P variants are common (5–10% in patients, 2–5% in controls) [49,50,51].

Recent reports suggest that loss of function variants in *LRRK2* are not associated with PD, arguing that haploinsufficiency is neither causative nor protective of PD [52].

#### 2.2.2. Pathophysiology

All the definite *LRRK2* mutations are in the catalytic domains and may result in hyperactivation of the kinase domain [3,53]. LRRK2 is involved in a large array of cell biological processes, and the disease mechanism may reflect important roles in microtubule function and Rab proteins as phosphorylation substrates [2,54].

### 2.3. VPS35

#### 2.3.1. Genotype-Phenotype

*VPS35* is implicated in autosomal dominant PD [55,56], with the missense variant p.D620N being the only mutation confirmed to date. This variant appears to be a mutational hotspot identified in different ethnic populations [57]. The mutation has an overall prevalence of 0.115% from the reported studies but may be as high as 1% in autosomal dominant PD [57,58,59]. The phenotype resembles idiopathic PD with a median age at onset of 49 years, levodopa responsiveness, and predominant tremor [5,58]. A recent study suggests that disease progression may be slow, with minimal cognitive impairment even after more than 10 years of disease onset [60].

#### 2.3.2. Pathophysiology

VPS35 plays a critical role in endosomal trafficking, but there is emerging evidence for a role in mitochondrial function [61]. The p.D620N mutation impairs the sorting function of the retromer complex, resulting in a disturbance of maturation of endolysosomes and autophagy, membrane receptor recycling, and mitochondrial-derived vesicle formation [2,59,62]. There may also be a role in neurotransmission and an interaction with other genes causing monogenic PD (such as *SNCA*, *LRRK2,* and *PRKN*) [62].

## 3. Autosomal Recessive Forms

### 3.1. PRKN

#### 3.1.1. Genotype-Phenotype

Mutations in *PRKN* are the most common cause of early-onset PD (EOPD), particularly in European populations. A recent study by Lesage and colleagues demonstrated that *PRKN* mutations account for 27.6% of autosomal recessive families [13]. They found that the proportion of probands with *PRKN* mutations is higher the younger the age at onset (AAO), as follows: 42.2% for AAO less than or equal to 20 years, 29% for 21 to 30 years, 13% for 31 to 40 years, but only 4.4% for 41 to 60 years [13].

A variety of different mutation types are described, including structural variants (43.2%, including exonic deletions, duplications, and triplications), missense mutations (22.3%), and frameshift mutations (16.5%) [14,63]. Deletions in exon 3 are the most common mutation [14]. Furthermore, a deletion of the *PRKN* and *PACRG* gene promoter has also been described in autosomal recessive PD [63].

PD-*PRKN* is characterized phenotypically by an early age at disease onset, lower limb dystonia at presentation, absence of cognitive impairment, a good and sustained response to levodopa, and frequent motor fluctuations and dyskinesias [64].

#### 3.1.2. Pathophysiology

Mutations in *PRKN* and *PINK1* likely disturb PINK1/parkin-mediated mitophagy, which is the selective degradation of mitochondria, a function essential for mitochondrial homeostasis [65]. In brief, parkin is a E3 ubiquitin ligase that ubiquinates outer mitochondrial membrane proteins such as mitofusin 1 and 2 [66]. PINK1 phosphorylates parkin and maintains its mitochondrial stabilization and translocation, mediating parkin activation [2,66].

### 3.2. PINK1

#### 3.2.1. Genotype-Phenotype

*PINK1* is the second most common cause of autosomal recessive PD and is characterized by typical Parkinson’s features such as tremor, bradykinesia, and rigidity, with a median age of onset of 32 [14,67]. Additional phenotypic features include dyskinesias in 39%, dystonia in 21%, and motor fluctuations in 34%, with cognitive impairment and psychosis occurring rarely (14% and 9%, respectively) [14]. The disease is slowly progressive, with a sustained response to levodopa therapy, although with an increased tendency for levodopa-induced dyskinesias.

The main mutation type was missense mutations (47.6%), then structural variants (19.1%), followed by nonsense mutations (14.3%) [14]. The most common specific mutation was a missense mutation, c.1040T>C (p.Leu347Pro) [14].

A recent paper suggests that the c.1040T>C mutation is frequently found in patients from the Pacific Islands [15]. The allele frequency was particularly high in West Polynesians (2.8%), which would translate to a homozygosity of 1 in 5000 people, suggesting that this could have a major contribution to EOPD in the region [15].

#### 3.2.2. Pathophysiology

See *PRKN* above.

### 3.3. PARK7

#### 3.3.1. Genotype-Phenotype

Mutations in *PARK7* can cause early-onset autosomal recessive parkinsonism, with at least 20 mutations in the *PARK7* gene identified. The majority of *PARK7* mutation carriers have EOPD (83%), whereas 13% have juvenile onset and 4% have late onset [14]. Recently, a Turkish family with juvenile PD was found to have a novel deletion of the neighboring genes of *PARK7* and *TNFRSF9*, raising the possibility of TNFRSF9 as a disease modifier [68].

#### 3.3.2. Pathophysiology

DJ-1 is ubiquitously expressed and is highly expressed in cells with high energy demands. DJ-1 exerts an antioxidative stress function through scavenging reactive oxygen species, regulation of transcription and signal transduction pathways, and acting as a molecular chaperone and enzyme [69]. Mutations within the *PARK7* gene substantially affect the survival of cells in oxidative environments, potentially leading to PD [70,71].

## 4. X-Linked Dystonia-Parkinsonism

### 4.1. Genotype-Phenotype

X-linked dystonia-parkinsonism (XDP), also referred to as Lubag, is a movement disorder initially described in Filipino males, caused by the insertion of a SINE-VNTR-Alu (SVA)-type retrotransposon in intron 32 of the *TAF1* gene [72,73]. The prevalence is 0.34 per 100,000 in the Philippines, with a high prevalence on the Island of Panay of 5.24 per 100,000 [74]. It initially presents with dystonia, and predominantly involves the craniocervical region that can become generalized at a later stage [72,75]. It may also present with parkinsonism, or this can develop later in the disease course [75]. Therefore, it can show longitudinal evolution from a hyperkinetic to a hypokinetic movement disorder. Although it primarily affects males, manifesting female carriers have been reported. The median age at onset is 40 years from a recent MDSGene review [72].

### 4.2. Pathophysiology

Recent evidence suggests that probands with XDP have reduced expression of the canonical TAF1 transcript [73]. De novo assembly of multiple neuronal lineages derived from pluripotent stem cells showed reduced expression due to alternative splicing and intron retention close to the SVA [73]. CRISPR/Cas 9 excision of the SVA was able to rescue TAF1 expression, providing evidence of abnormal transcription mediated by the SVA in the pathophysiology of XDP [73]. Further evidence suggests that a hexanucleotide repeat within the SVA modifies disease expressivity, with the number of repeats showing an inverse correlation with the age at onset [76].

## 5. Complex or Atypical Forms

### 5.1. ATP13A2

#### 5.1.1. Genotype-Phenotype

Biallelic mutations in *ATP13A2* have been found to cause a complex form of parkinsonism known as Kufor-Rakeb syndrome (KRS), characterized by juvenile onset parkinsonism, cognitive impairment, and a supranuclear gaze palsy. *ATP13A2* mutations can also cause a range of phenotypes, including neuronal ceroid lipofuscinosis, hereditary spastic paraplegia, and juvenile amyotrophic lateral sclerosis.

Recently, perhaps the first postmortem study of KRS was reported [77]. This showed accumulation of lipofuscin in the neurons and glia, but an absence of Lewy body pathology as well as alpha-synuclein, TDP43, tau, and beta amyloid pathology. This provides evidence for a pathological link with neuronal lipofuscinosis rather than the typical findings in PD [77].

#### 5.1.2. Pathophysiology

*ATP13A2* mutations impair lysosomal and mitochondrial function. The mechanism may involve impaired lysosomal polyamine transport resulting in lysosome-dependent cell death [78].

### 5.2. DCTN1

#### 5.2.1. Genotype-Phenotype

*DCTN1*-associated Parkinson-plus disorder, also called Perry syndrome, is a rare autosomal dominant disorder characterized by rapidly progressive parkinsonism, depression and mood changes, weight loss, and progressive respiratory changes, chiefly tachypnoea and nocturnal hypoventilation [79].

The disease is linked to mutations in exon 2 of the *DCTN1* gene. The mean age at onset of disease is 48 years (range: 35–61) and the mean duration to death is 5 years since diagnosis, from either respiratory failure, sudden unexplained death, or suicide [80]. *DCTN1* mutations have been associated with additional phenotypes, including distal spinal and bulbar muscular atrophy and amyotrophic lateral sclerosis.

#### 5.2.2. Pathophysiology

*DCTN1* encodes p150glued, the major subunit of the dynactin complex which binds to the motor protein dynein which binds directly to microtubules and different dynactin subunits [80]. Mutations in *DCTN1* diminish microtubule binding and lead to intracytoplasmic inclusions [81].

### 5.3. DNAJC6

#### 5.3.1. Genotype-Phenotype

Biallelic mutations in *DNAJC6* cause juvenile-onset, atypical parkinsonism with onset during childhood and a very rapid disease progression with loss of ambulation within 10 years from onset [82,83]. Patients are poorly responsive to levodopa therapy and have additional manifestations such as developmental delay, intellectual disability, seizures, and other movement disorders (e.g., dystonia, spasticity, myoclonus). A minority of patients have early-onset parkinsonism, with symptom onset in the third to fourth decade of life and an absence of additional features [84]. These patients generally have a slower rate of disease progression and a favorable response to levodopa therapy.

#### 5.3.2. Pathophysiology

*DNAJC6* encodes for auxilin 1, a brain-specific form of auxilin and a co-chaperone protein involved in the clathrin-mediated synaptic vesicle endocytosis. Auxilin deficiency has been found in animal models to result in impaired synaptic vesicle endocytosis, and thus negatively impacts synaptic neurotransmission, homeostasis, and signaling [85]. However, the exact mechanism by which auxilin deficiency leads to dopaminergic neurodegeneration and atypical neurological symptoms remains unclear.

### 5.4. FBXO7

#### 5.4.1. Genotype-Phenotype

Mutations in *FBXO7* cause autosomal recessive, juvenile/early-onset parkinsonian-pyramidal syndrome (also called PARK15). Missense, splice site, and nonsense mutations have been reported. The median age at onset was 17 years, with a range of 10 to 52 years. The typical presenting symptoms were bradykinesia and tremor, and patients affected by this atypical parkinsonism frequently show pyramidal signs, dysarthria, and dyskinesia. Psychiatric manifestations, such as visual hallucination, agitation, aggression, disinhibition, and impulsive control disorder, are prominent in these patients as a complication of dopaminergic therapy [86,87,88,89].

#### 5.4.2. Pathophysiology

FBXO7 is expressed in various tissues, including the gray and white matters of the brain. It directly interacts with PINK1 and parkin to engage in mitophagy [90]. The loss of *FBXO7* expression has been shown to lead to a significant inhibition of parkin recruitment to depolarized mitochondria [90].

### 5.5. PLA2G6

#### 5.5.1. Genotype-Phenotype

Mutations in *PLA2G6* have been linked to a variety of neurological disorders, including infantile neuroaxonal dystrophy, neurodegeneration with brain iron accumulation 2B, and Karak syndrome. *PLA2G6* mutations may also result in another phenotype—autosomal recessive, adult-onset dystonia-parkinsonism (also called PARK14) [91].

Patients with *PLA2G6*-related parkinsonism first show symptoms in their childhood or early adulthood, with an age at onset ranging from 8 to 36. In addition to parkinsonism, the majority have dystonia [92,93]. Neuropsychiatric presentations such as depression, psychosis, and cognitive decline are common. There is a good response to levodopa therapy. Magnetic resonance imaging of the brain in most patients showed an absence of iron deposition, and if iron was present, it was found in the substantia nigra or globus pallidus, or both [94].

#### 5.5.2. Pathophysiology

PLA2G6, a phospholipase 2, catalyzes the hydrolysis of the sn-2 acyl-ester bonds in phospholipids to form arachidonic acid and other fatty acids. This is involved in the phospholipid remodeling, apoptosis, and prostaglandin and leukotriene synthesis. The exact mechanism of *PLA2G6* in neurodegenerative diseases remains obscure, however defective phospholipases have been implicated in the pathogenesis of neurodegenerative conditions with iron dyshomeostasis.

### 5.6. SYNJ1

#### 5.6.1. Genotype-Phenotype

Mutations in *SYNJ1* are linked to autosomal recessive, early-onset Parkinson disease-20 (PARK20). Individuals affected by *SYNJ1*-associated parkinsonism generally show symptoms in the third decade of life, and manifest parkinsonism (tremor, bradykinesia) with a poor response to levodopa treatment, as well as additional atypical signs such as dystonia, seizures, cognitive impairment, and developmental delay [95].

#### 5.6.2. Pathophysiology

Synaptojanin-1 plays a crucial role in synaptic vesicle dynamics, including endocytosis and recycling. SJ1-knockout mice display endocytic defects and a remarkable accumulation of clathrin-coated intermediates [96]. Fasano et al. further showed that SYNJ1 is critically involved in early endosome function, and that a loss of *SYNJ1* leads to impaired recycling of the transferrin receptor to the plasma membrane, highlighting the important role that the autophagy-lysosome pathway plays in PD pathogenesis [92].

## 6. Recently Described Parkinson’s Disease Genes

### 6.1. CHCHD2

#### 6.1.1. Genotype-Phenotype

Mutations in the *CHCHD2* gene were linked to an autosomal dominant, late-onset form of PD (PARK22) in the Japanese population in 2015 by Funayama et al., who reported two missense mutations (p.T61I, p.R145Q) and a splice-site mutation (c.300+5G>A) in the *CHCHD2* gene [93]. Both missense mutations were also reported in the Chinese population [97,98], although were not found in a study on a large cohort of PD patients of western European ancestry [99]. Instead, three rare variants (p.A32T, p.P34L, and p.I80V) in the *CHCHD2* gene were found in the western European cohort, occurring in highly conserved residues [99]. A homozygous missense mutation (p.A71P) has also been reported in a 26-year-old Caucasian woman with recessive early-onset PD [100]. Patients affected by *CHCHD2*-associated PD typically present with typical parkinsonian features, with a significant response to levodopa.

#### 6.1.2. Pathophysiology

CHCHD2 contains a mitochondrial-targeting sequence at the N-terminus and localizes to the mitochondrial intermembrane space. Its close homologue CHCHD10 is enriched at crista junctions of the mitochondria and is believed to be involved in oxidative phosphorylation or in maintenance of crista morphology [101]. The loss of CHCHD2 in flies leads to mitochondrial and neural phenotypes associated with PD pathology and causes chronic oxidative stress and thus age-dependent neurodegeneration in the dopaminergic neurons [102].

### 6.2. LRP10

#### 6.2.1. Genotype-Phenotype

Through genome-wide linkage analysis of an Italian family with autosomal dominant PD, Quadri and colleagues implicated the *LRP10* gene on chromosome 14 as a possible causative disease gene [31]. This was verified through analysis of a larger cohort of patients, where rare, potential mutations in *LRP10* were found to be associated with PD and dementia with Lewy bodies [31]. These findings were unable to be replicated in a study by Tesson et al., whose co-segregation analysis did not support a causal role for *LRP10* in PD [103]. Since then, several additional variants in the *LRP10* have been identified in patients with PD, progressive supranuclear palsy, frontotemporal dementia, and amyotrophic lateral sclerosis, although the correlation of *LRP10* variants with the development of α-synucleinopathies and other neurodegenerative diseases has been debated [104,105,106].

#### 6.2.2. Pathophysiology

LRP10 is a single-pass transmembrane protein and a member of a subfamily of LDL receptors. Grochowska et al. discovered that *LRP10* expression was high in non-neuronal cells but undetectable in neurons, and that it was present in the trans-Golgi network, plasma membrane, retromer, and early endosomes in astrocytes [107]. They suggested that LRP10-mediated pathogenicity involves the interaction of LRP10 and SORL1 in vesicle tracking pathways, as they were shown to co-localize and interact, and that disturbed vesicle trafficking and loss of LRP10 function were crucial in the pathogenesis of neurodegenerative diseases [107].

### 6.3. TMEM230

#### 6.3.1. Genotype-Phenotype

The link with PD was first proposed in 2016 by Deng et al., who investigated a large Canadian Mennonite pedigree with autosomal dominant, typical PD, and discovered a p.R141L mutation in *TMEM230* which reportedly fully co-segregated with disease [34]. The same pedigree was investigated by Vilarino-Guell and colleagues, who identified a heterozygous missense variant in *DNAJC13* (p.N855S) which did not fully co-segregate with disease [108]. Whilst *TMEM230* variants have been identified in further studies on PD patient groups, other follow-up genetic studies have failed to detect PD-associated *TMEM230* variants, and whether evidence exists for ‘proof of pathogenecity’ has been debated [109,110].

#### 6.3.2. Pathophysiology

TMEM230 is a transmembrane protein with ubiquitous expression. It is a trafficking protein of secretory and recycling vesicles, including neuronal synaptic vesicles. Expression of mutant *TMEM230* was found to lead to increased α-synuclein levels [34]. Loss of function of TMEM230 impairs secretory autophagy, Golgi-derived vesicle secretion, and retromer trafficking [111].

### 6.4. UQCRC1

#### 6.4.1. Genotype-Phenotype

An association between *UQCRC1* mutations and familial PD was first reported by Lin et al. in 2020, who identified a novel heterozygous mutation (p.Y314S) in the *UQCRC1* gene which co-segregated with disease in a Taiwanese family with autosomal dominant parkinsonism with polyneuropathy [112]. An additional variant in *UQCRC1* (p.I311L) also co-segregated with disease [112]. In a subsequent study, no common variant was found to be significantly associated with PD in the European population [113].

#### 6.4.2. Pathophysiology

UQCRC1 is a core component of complex III in the respiratory chain. In Drosophila and mouse models, URCRC1 p.Y314S knock-in organisms showed dopaminergic neuronal loss, age-dependent locomotor deficits, and peripheral neuropathy [112]. Disruption of the *Uqcrc1* gene in mice causes embryonic lethality [114], and deficiency of Uqcrc1 in Drosophila increases the cytochrome c in the cytoplasmic fraction and activates the caspase cascade, thus causing a reduction of dopaminergic neurons and neurodegeneration [115].

### 6.5. VPS13C

#### 6.5.1. Genotype-Phenotype

Lesage et al. first reported five truncating mutations in *VPS13C* in three unrelated PD patients [35]. These probands were either homozygous or compound heterozygous and had a distinct phenotype of EOPD which progressed rapidly and showed a good but transient initial response to levodopa treatment. Additional variants in *VPS13C* have been identified in further reports on autosomal recessive, early-onset forms of parkinsonism, although not in late-onset PD [116].

#### 6.5.2. Pathophysiology

VPS13C is part of the family of conserved VPS13 proteins and behaves similarly to VPS35 (see above). VPS13C is a phospholipid transporter and localizes to the contact sites between the endoplasmic reticulum (ER) and late endosome [117]. VPS13 proteins are thought to mediate endoplasmic reticulum-phagy at late endosomes [117].

## 7. Rare, Atypical, and Unconfirmed Forms

There are many genes that can cause parkinsonian phenotypes, and comprehensive lists can be found elsewhere, with over 70 different genes causing early-onset parkinsonism or parkinsonism as part of a complex neurological disorder [118]. Clinicians should be especially vigilant for treatable causes such as Wilson’s disease [118]. Mutations in *GCH1* can cause dopa-responsive dystonia and PD and should also be considered. *POLG* mutations can cause movement disorders including parkinsonism and dystonia. Mutations in *PTRHD1* can cause autosomal recessive PD with intellectual impairment but are rare [119]. *RAB39B* mutations can cause X-linked intellectual impairment and parkinsonism with classic Lewy body pathology on autopsy studies [120]. Several additional reported genes have not been independently replicated and perhaps require further validation before being considered PD genes, such as *DNAJC13*, *EIF4G1*, *GIGYF2*, *HTRA2*, and *UCHL1* [121].

## 8. Risk Variants versus Monogenic Forms

When discussing genetic risk in PD, one should differentiate risk variants from causative monogenic ones. Risk variants are relatively common, each with an individual small effect size, yet collectively they significantly increase disease risk. A recent large meta-analysis of genome-wide association studies (GWAS) identified 90 such genome-wide risk alleles that collectively account for 16–36% of PD heritability [122]. A causative monogenic variant, on the other hand, is a rare variant with a large effect size, that is considered the causative culprit of the disease. Complicating this oversimplified dichotomic differentiation is the fact that autosomal dominant forms of monogenic PD have incomplete age-dependent penetrance to a variable extent, which may be affected by the causative gene and the specific pathogenic variant as well as the patient’s ethnicity. Moreover, a complex interplay between monogenic causative variants and risk variants may affect disease penetrance, as exemplified by a recent study which showed that disease penetrance of the *LRRK2* variant p.G2019S is modified by a polygenic risk score [45].

## 9. *GBA* Variants

A notable issue is the one related to pathogenic *GBA* (or *GBA1*) variants, which constitute the most common genetic risk factor for PD. These variants are found in approximately 8.5% of PD patients [123]. However, this number varies significantly across different ethnic groups, ranging between 2.3% and 12% in populations of non-Ashkenazi Jewish origin to 10–31% in Ashkenazi Jews [124]. *GBA* variants were more common in patients with early-onset disease (<50 years), more rapid development of dementia, and a more aggressive motor course [125,126]. Pathogenic variants in this gene have a low, age-dependent penetrance in PD, which is highly variable across different reports, ranging between 8% and 30% by age 80 years [127,128,129,130]. In a recent study, the authors used a kin-cohort design to evaluate the penetrance of pathogenic *GBA* variants in a cohort of unselected PD patients, showing that the risk to develop disease by age 60, 70, and 80 years was 10%, 16%, and 19%, respectively [131]. This study also found a trend towards a greater PD penetrance for severe pathogenic variants compared to mild pathogenic variants in the *GBA* gene, although this difference did not reach statistical significance [131].

Adding to the complexity of *GBA*-associated PD, a recent study demonstrated an association between PD polygenic risk score and both penetrance and age at onset in individuals carrying a disease-associated *GBA* variant [132]. Another study examined PD clustering in eight families of non-Parkinsonian *GBA*-p.N370S homozygote Gaucher patients, showing that all PD cases in these families stemmed from only one of the proband’s parents, further highlighting the potential role of genetic modifiers in PD risk among carriers of *GBA* variants [133].

Furthermore, a recent study showed that both pathogenic (i.e., associated with Gaucher disease) and non-pathogenic (i.e., not associated with Gaucher) variants in *GBA* are common in PD, with a more aggressive course in terms of dementia and motor progression [126].

In summary, *GBA* variants are a common risk factor for PD. They should be clearly differentiated as such from monogenic causes for PD, to avoid ambiguity and terminological and conceptual perplexity when discussing PD risk with patients and clinicians.

## 10. Genetic Testing in Parkinson’s Disease

Genetic workup is not routinely performed as part of PD evaluation, and movement disorder specialists only very occasionally suggest genetic testing to PD patients. This is due to a combination of factors related to cost, lack of physician’s perceived impact on patient’s management, and physician’s discomfort regarding test selection and its results or their impact on the patients and their family members [134]. The field of genetic testing in PD is rapidly evolving during recent years, due to the better availability of next-generation sequencing (NGS)-based molecular tests and the initiation of genetic diagnosis-based interventional clinical trials.

### 10.1. Who Should Be Offered Genetic Testing in Parkinson’s Disease?

Traditionally, a monogenic cause would most probably be suspected, and therefore a genetic test considered, in patients with early-onset PD before age 50 years, and particularly before age 40 years. Furthermore, although polygenic risk and multifactorial inheritance would probably explain most cases with familial clustering of PD, a striking familial history, either of autosomal dominant or autosomal recessive pattern, is yet another clue for a possible monogenic cause that may suggest that a genetic test should be considered. Ethnic origin may also affect the decision to perform genetic testing, for example in patients of Ashkenazi Jewish or African Berber origin. As opposed to this traditional case-by-case approach, as molecular testing is becoming more available, a recently suggested permissive approach supports a more widespread use of genetic testing in PD to improve patient care, to allow inclusion of patients in molecular diagnosis-based clinical trials, and to benefit therapeutic insights and strategies for the larger PD population, including patients with idiopathic disease [135]. This notion can tremendously benefit PD patients both individually and collectively. However, it should be backed up by thorough knowledge of the different evolving aspects of genetic testing in PD, and by an individually tailored explanation to patients and potential carriers in their family prior to testing as well as when returning them the test results, regarding the test and the potential implications of its results for them and for their family members.

### 10.2. The Implications of a Genetic Diagnosis in Parkinson’s Disease

A genetic diagnosis may have significant implications for PD patients, both for expected disease course and response to therapeutic interventions. As mentioned, several monogenic forms are expected to respond well to levodopa medication (e.g., *PRKN*), whereas others are poorly responsive (e.g., *DNAJC6*) (Table 2). Additionally, a recent study found that the rate of cognitive decline for *GBA* mutation carriers after bilateral subthalamic nucleus deep brain stimulation (STN-DBS) is higher than that of carriers of *PRKN* and *LRRK2* mutations and those without identified disease-associated pathogenic variants [136] (Figure 1A). These findings were further corroborated by a new study which suggests that STN-DBS is associated with a greater rate of cognitive decline in *GBA* mutation carriers [137]. A recommendation that arose from this study is that PD patients should be screened for *GBA* pathogenic variants prior to DBS surgery, and that carriers of such variants should be counseled on the greater risk of cognitive decline [137].

For *SNCA*-PD, the response to DBS may also differ according to the type of mutation (Figure 1B). A recent report of four patients with *SNCA* mutations showed a good response in the three patients with duplications and a poor response in the patient with a missense mutation (p.A53E) [138] (Table 1).

In addition to implications for DBS, the emerging importance of a genetic diagnosis in PD is also related to new gene-based targeted approaches that are being developed in recent years [3], since a specific molecular genetic diagnosis may allow inclusion in interventional clinical trials that target a genetically determined subgroup of PD patients. Moreover, a genetic diagnosis for additional family members at risk of developing PD allows for a more accurate estimation of recurrence risk and informs genetic counseling and family planning. Moreover, some patients are greatly distressed just by the uncertainty regarding the cause for their condition and a genetic diagnosis may bring them great relief.

### 10.3. Challenges in Genetic Testing

The challenges in genetic testing in PD are related to the patient, the choice of genetic test, and the test results. Patients may be reluctant to perform genetic testing due to different reasons, including a lack of perceived benefit, concern regarding the implications of the test results for them or their family members, or cost. Genetic counseling prior to performing a genetic test is non-directive, meaning that patients or their relatives cannot be directed to have a genetic test, however it should include a thorough, individually tailored explanation regarding the reason why a genetic test is offered, the test itself, its advantages and limitations, and the potential implications of the test results for the patient and their family members. This type of pre-test discussion with the patient is necessary to address the patient’s concerns and to ensure that they are given all the required information to make a knowledge-based decision on whether to proceed with genetic testing or not.

Many types of genetic tests are available in clinical and research settings, ranging from focused testing for a single gene or a specific variant, through variant panels and gene panels, to exome or genome sequencing. Due to the increase in availability and decrease in cost of NGS-based tests, the traditional approach of testing one gene at a time was largely replaced in recent years with broader tests, such as gene panels and exome or genome sequencing, except when a known pathogenic variant has been previously found in the patient’s family, or in uncommon cases where a very high suspicion is raised for a specific gene. When choosing to use a gene panel, one should consider the considerable variability in gene content of different panels. A recent study evaluated the types of clinical genetic tests that are used in PD, revealing notable differences in gene panel size, ranging from 5 to 62 genes. That study showed that five genes were included in all panels (*SNCA*, *PRKN*, *PINK1*, *PARK7* (*DJ1*), and *LRRK2*), while *VPS35* and *GBA* were only variably included, and that the differences between panels were mainly the result of the variable inclusion of genes associated with atypical parkinsonism and dystonia disorders, or genes with an uncertain association with PD [139]. The selected gene panel should ideally include all established genes for PD with both sequence and deletion/duplication analysis. In cases where the patient presents a combined or an atypical phenotype, a broader approach should be considered, either by using a more comprehensive gene panel or by a genomic analysis with exome or genome sequencing, depending on the specific clinical indicators. Notable limitations that should be taken into consideration are the ones associated with the *GBA* gene, for which a related pseudogene and structural variations may complicate the detection of pathogenic variants. A novel approach is to use long-read sequencing to assess this gene, with the GridION nanopore sequencing platform recently used in a New Zealand cohort of patients [139]. Another factor to consider is the cost of genetic tests, which might not be covered by the patient’s insurance and therefore may inevitably affect decisions in the molecular workup in some cases. In summary, the decision regarding which genetic test should be used depends on case-specific factors and requires to consider the different types of tests available, their advantages and limitations, and their suitability for each individual patient.

## 11. Role for Heterozygosity in Autosomal Recessive Parkinson Genes

The possibility that monoallelic pathogenic variants in autosomal recessive PD genes constitute a risk factor for PD is controversial, and conflicting evidence regarding this issue has been reported.

### 11.1. PRKN Heterozygotes

A recent population-based study analyzed data of 164 confirmed heterozygous *PRKN* mutation carriers and 2582 controls from South Tyrol in Northern Italy. This study showed a significantly higher number of carriers than controls with a reported akinesia-related phenotype based on a validated PD screening questionnaire [140]. Another study evaluated *PRKN* as a risk factor for PD in three large independent case-control cohorts and revealed a 1.55-fold risk increase in heterozygous carriers, who also had a younger age of disease onset [141]. However, ~70% of potentially monoallelic cases were not assessed for a second *PRKN* mutation. To further address this, the authors conducted a meta-analysis of available cohorts and studies of individuals from European ancestry, demonstrating a significant 1.65-fold increase in PD risk in monoallelic *PRKN* mutation carriers. Nevertheless, when excluding from the analysis studies which did not search for biallelic carriers and those that focused on early-onset PD, no association between monoallelic *PRKN* mutation and disease risk was found, highlighting the importance of confounding factors that might bias this association [141]. In a recent study, full sequencing and CNV analysis of *PRKN* in 2809 PD patients and 3629 controls revealed no association between all types of heterozygous *PRKN* variants and PD risk [142].

### 11.2. PINK1 Heterozygotes

Several studies have previously suggested that heterozygous *PINK1* variants may act as a risk factor for late-onset PD. Of note, one study in a large German family suggested that heterozygous *PINK1* mutations may increase the risk for the development of at least subtle motor and non-motor signs of PD [143]. Puschmann et al. investigated the functional effects of the heterozygous *PINK1* p.G411S variant and concluded that it acts as a risk factor for PD, which confers its effect by a partial dominant-negative mechanism [144]. A recent comprehensive analysis contradicted these studies. By harnessing combined data from several large datasets totaling 13,708 cases and 362,850 control individuals, this investigation found no evidence of association between heterozygous *PINK1* mutations and PD risk [145], further highlighting the complexity and controversy in this field.

### 11.3. Conclusion on Heterozygous Carriers

The evidence for the role of heterozygous carriers is conflicting—some studies which were based largely on findings in specific cases or families suggested a possible association, while newer studies that utilized large datasets mostly refuted this possibility.

A hidden trans-acting pathogenic variant on the other allele of the gene may at least partly explain these contradictory findings. This may occur in cases where the chosen methodology could not identify these variants, for example when a deletion/duplication analysis was not performed or when the second allele harbored a disease-associated non-coding, structural, or mosaic variant which the molecular testing strategy that was used could not reveal. In these cases, an apparent association between a monoallelic variant and disease may be erroneously concluded. This scenario, however, would not explain cases of families with a clear autosomal dominant inheritance pattern across several generations. Another possible explanation for the conflicting evidence may be that monoallelic deleterious variants in autosomal recessive Parkinson-related genes confer an increased disease risk to some extent as part of a multifactorial inheritance, where each individual, family, or ethnic group are affected by a certain genetic background and/or environmental factors. In this potential scenario, while a monoallelic pathogenic variant may indeed increase the risk for PD, the threshold for disease expression may vary substantially between different individuals, families, or ethnic groups, depending on other genetic variants and environmental factors. This might be missed when analyzing very large, grouped datasets or data that are limited to specific ethnic groups. Other potential factors that might contribute to those contradictory findings may stem from data collection-related biases, such as a recall bias or cases of subtle signs of parkinsonism in reportedly healthy individuals which are considered in the analysis as unaffected controls.

## 12. Dual LRRK2 and GBA Mutation Carriers

It would be anticipated that having a mutation in both *LRRK2* and *GBA* would have an added deleterious effect, as suggested by laboratory studies [146,147]. However, a recent longitudinal study of a large PD sample measuring progression using the Montreal Cognitive Assessment and Movement Disorders Society—Unified Parkinson Disease Rating Scale–Part III, showed that patients with both the p.G2019S mutation and *GBA*-PD had a slower rate of decline than those with *GBA*-PD alone, which was no different from *LRRK2*-G2019S alone [148]. Similarly, a retrospective observational study of Ashkenazi Jewish patients revealed that patients with mutations in *LRRK2* and *GBA* (described by the authors as “*GBA*-*LRRK2*-PD”) were less frequently affected by dementia, probable REM-behavior sleep disorder, and psychosis, compared to other groups (*GBA*-PD, *LRRK2*-PD, mutation-negative PD) [149]. This raises the possibility of a protective effect of having the *LRRK2* p.G2019S mutation in *GBA* mutation carriers [149].

## 13. Conclusions

There have been major advances in research into monogenic PD in recent years. There have been multiple PD gene discoveries, although we highlight the importance of independent validation of these findings. There have been greater insights into genotype–phenotype relationships, and laboratory studies have translated the genetic discoveries into an improved understanding of the pathophysiological mechanism underlying PD.

It has become apparent that there are major ethnic and regional differences in the distribution of mutations in PD genes. There has been further evidence on the role of heterozygous carriers in autosomal recessive PD genes, and the effect of having mutations in both *LRRK2* and *GBA* in the same individual. Additionally, there is a suggestion that the underlying monogenic cause may influence the disease course as well as the response to levodopa and DBS.

Advances in genomic technology provide individuals with PD with greater access to genetic testing through both clinical and research pathways. Global efforts will play a key role in exploiting this genomic data. Worldwide studies can pool many patients to identify rare genetic causes of PD and can also be used to attempt to replicate important genetic discoveries. Furthermore, they offer greater representation of underrepresented populations from different ethnic groups and geographic regions. There are several major global projects to identify new disease genes in PD, including established initiatives such as the International Parkinson Disease Genomics Consortium [150] and newer initiatives such as the Global Parkinson’s disease genetics program (GP2) [151].

PD currently remains an incurable disorder but advances in our understanding of the genetics of PD may inform our understanding of the pathophysiology and thus help with efforts to develop targeted therapies.

## Figures and Tables

**Figure 1 genes-13-00471-f001:**
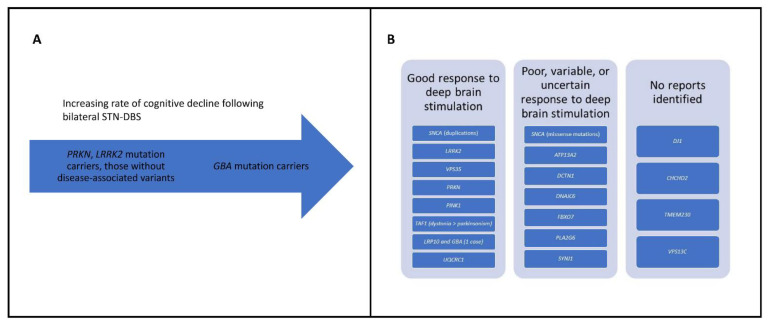
(**A**) Increasing cognitive decline in *GBA* carriers versus *PRKN*, *LRRK2*, and those without disease-associated variants. (**B**) Outcome of deep brain stimulation stratified according to Parkinson’s disease monogenic forms.

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
