# Peer review of "Monogenic Parkinson’s Disease: Genotype, Phenotype, Pathophysiology, and Genetic Testing"

_genes, 2022, doi:10.3390/genes13030471_

Round 1

Reviewer 1 Report

Dear Authors,

Following the analysis of the manuscript titled " Monogenic Parkinson's Disease- Genotype, Phenotype, Patho-2 physiology, and Genetic Testing", I find the topic of the article is intriguing, the content is presented comprehensively, and such detailed review can provide evidence for the assertions made. I recommend revising it with these observations in mind:

Point 1: I recommend that the authors provide an index at the beginning of their lengthy manuscript to help guide readers.

Point 2: The bibliography needs to be updated because more than 20 percent of the articles have been published for more than a decade. In addition, the reference list was excessively long.

Point 3: Since figures 1 and 2 are in fact tables, I request that the authors make them available as tables.

Point 4: The authors went into great detail about each genotype's response to levodopa treatment, but they did not draw any conclusions about the most commonly prescribed anti-Parkinson agent. I recommend the authors to conclude their remarks in this regard.

Point 5: I encourage authors to include a brief note in the conclusion section about how monogenic PD impacts current management of the disease.

Author Response

Following the analysis of the manuscript titled " Monogenic Parkinson's Disease- Genotype, Phenotype, Patho-2 physiology, and Genetic Testing", I find the topic of the article is intriguing, the content is presented comprehensively, and such detailed review can provide evidence for the assertions made. I recommend revising it with these observations in mind: 

Point 1: I recommend that the authors provide an index at the beginning of their lengthy manuscript to help guide readers.

Authors’ response:

Thank you, we contacted the journal who informed us that all published papers have an index of content on the left side of the web page which makes it easier for readers to scroll through the paper.

Point 2: The bibliography needs to be updated because more than 20 percent of the articles have been published for more than a decade. In addition, the reference list was excessively long.

Authors’ response:

Thank you for the suggestion, we have cut down the reference list substantially with a focus on removing older articles. The number of references has gone from 190 to 150. In addition, we have shortened the manuscript overall.

Point 3: Since figures 1 and 2 are in fact tables, I request that the authors make them available as tables.

Authors’ response:

Thank you for the suggestion, we have converted figure 1 to a table.

We have added a panel to figure 2 (panel 1), and believe this is more consistent with a figure, and makes our point with greater visual impact:

“Figure 1. Panel A. Highlights increasing cognitive decline in GBA carriers versus PRKN, LRRK2, and those without disease-associated variants. Panel B. Outcome of deep brain stimulation stratified according to Parkinson’s disease monogenic forms.”

Point 4: The authors went into great detail about each genotype's response to levodopa treatment, but they did not draw any conclusions about the most commonly prescribed anti-Parkinson agent. I recommend the authors to conclude their remarks in this regard.

Authors’ response:

We thank the reviewer for the suggestion, in the section on the implications of a genetic diagnosis, we have now included the following sentence:

“As mentioned, several monogenic forms are expected to respond well to levodopa medication (e.g. PRKN) whereas others are poorly responsive (e.g. DNAJC6) (Table 2).”

Point 5: I encourage authors to include a brief note in the conclusion section about how monogenic PD impacts current management of the disease.

Authors’ response:

We agree with this suggestion, and have added the following sentence to the conclusion:

“Furthermore, there is a suggestion that the underlying monogenic cause may influence the disease course as well as the response to levodopa and DBS outcomes.”

Reviewer 2 Report

The manuscript is sufficiently clear, some improvements in figures are required (please improve table and provide chart i.e. pie chart or others to be  of quick understanding). It would be appreciable a more extensive discussion   on the implication of the study.

Author Response

The manuscript is sufficiently clear, some improvements in figures are required

Point 1: Please improve table and provide chart i.e. pie chart or others to be  of quick understanding.

Authors’ response:

We have made changes to Table 1 by shortening the content. We converted Figure 1 into a table, as suggested by Reviewer 1. We adjusted Figure 2 (now figure 1), added an extra panel (see response to Reviewer 1). We considered a pie chart but thought perhaps it would not be relevant given there are no numerical values in the table.

Point 2: It would be appreciable a more extensive discussion on the implication of the study.

Authors’ response:

Please see response to Reviewer 1 when we discuss the therapeutic implications in more detail. E.g. we added the following sentences:

“As mentioned, several monogenic forms are expected to respond well to levodopa medication (e.g. PRKN) whereas others are poorly responsive (e.g. DNAJC6) (Table 2).”

“Furthermore, there is a suggestion that the underlying monogenic cause may influence the disease course as well as the response to levodopa and DBS outcomes.”

Reviewer 3 Report

This review by Jia et al. summarizes and discusses the monogenic forms of Parkinson’s disease with a focus on the clinical implications of genotype, phenotype, pathophysiology and geographic and ethnic distribution. It highlights that monogenic Parkinson’s disease is influenced by ethnicity and geographical differences.

This is a review with a lot of work but with some criticisms, mainly related with the genes included in this review.

Why authors have included some genes such as DCTN1 (related with atypical PD phenotype) but don’t others such as POLG?

In my opinion a little description of each gene (an idea of the role of the protein coding by each gene) should be provided before talk about genotype-phenotype. Other option would be talk about pathophysiology before genotype-phenotype.

Table 1 must be reviewed:

I suggest to change “white” term and use other such as Caucasian.

The width of the columns should be revised to better fit the words.

Most frequent mutation is described in some genes but doesn’t in others. In addition, those descriptions of “most common mutation” are in different columns.

There are some unclear sentences. For example: “So far there have been 8 missense variants identified as causing autosomal dominant PD: p.A30G, p.A30P, p.E46K, p.H50Q, p.G51D, p.A53E, p.A53T, and p.A53V. An MDSGene review identified phenotypic differences between these missense variants, with the commonest mutation p.A53T having an early age at onset compared to p.A30P and p.E46K [7]”. Authors say “these missense variants” (all of them) but only referee differences bamong p.A53Tand p.A30P and p.E46K. Although some of them are subsequently discussed,  What about the others?.

Table 1 and Figure 1 give a bit repetitive information.

Genes’ names must be in cursive.

Some of the “Recently described Parkinson’s disease genes” are not so recent.

In X-linked Dystonia Parkinsonism. Reader doesn´t know what gene authors are talking about until TAF1 is named in pathophysiology.

Why authors talk about “Dual LRRK2 and GBA mutation carriers” so separately from the other genetic information?

In “Conclusion”, authors repeat the conclusions from other works.

“GBA variants” must be reviewed. It has been shown that variants in GBA affect the phenotype of PD, but not only pathogenic variants.

Author Response

This review by Jia et al. summarizes and discusses the monogenic forms of Parkinson’s disease with a focus on the clinical implications of genotype, phenotype, pathophysiology and geographic and ethnic distribution. It highlights that monogenic Parkinson’s disease is influenced by ethnicity and geographical differences.

This is a review with a lot of work but with some criticisms, mainly related with the genes included in this review.

Point 1: Why authors have included some genes such as DCTN1 (related with atypical PD phenotype) but don’t others such as POLG?

Authors’ response:

We haven’t been able to cover all the genes causing parkinsonism, as mentioned in the text. We focused on monogenic forms where parkinsonism is the common and predominant feature. However, we have now mentioned POLG in the rare forms section. We have also referenced a paper by Morales-Briceno and colleagues who list all the genes related to Parkinson’s disease/parkinsonism in great detail.

Point 2: In my opinion a little description of each gene (an idea of the role of the protein coding by each gene) should be provided before talk about genotype-phenotype. Other option would be talk about pathophysiology before genotype-phenotype.

Authors’ response:

We thank the reviewer for this suggestion. We tested both ways, but felt it worked better to discuss the genotype-phenotype first, then the pathophysiology, given that the gene discovery and phenotypic characterisation often occurs first, followed by the detailed laboratory studies exploring the underlying genetic mechanisms.

Point 3: Table 1 must be reviewed:

I suggest to change “white” term and use other such as Caucasian.

The width of the columns should be revised to better fit the words.

Most frequent mutation is described in some genes but doesn’t in others. In addition, those descriptions of “most common mutation” are in different columns.

Authors’ response:

In response to these suggestions:

  • We changed the term “white” to “European”.
  • The width of the columns was revised to better fit the words.
  • For simplicity, we have removed mention of the most common mutations from Table 1.

Point 4: There are some unclear sentences. For example: “So far there have been 8 missense variants identified as causing autosomal dominant PD: p.A30G, p.A30P, p.E46K, p.H50Q, p.G51D, p.A53E, p.A53T, and p.A53V. An MDSGene review identified phenotypic differences between these missense variants, with the commonest mutation p.A53T having an early age at onset compared to p.A30P and p.E46K [7]”. Authors say “these missense variants” (all of them) but only referee differences among p.A53Tand p.A30P and p.E46K. Although some of them are subsequently discussed,  What about the others?

Authors’ response:

Given that there are few reports of these rare missense variants, it is hard to draw conclusions.

We have clarified the sentence by changing it to:

“An MDSGene review identified phenotypic differences between some of these missense variants, with the commonest mutation p.A53T having an early age at onset compared to p.A30P and p.E46K [5]. However, these findings are uncertain given that the number of cases for SNCA missense variants other than p.A53T is small [5].”

Point 5: Table 1 and Figure 1 give a bit repetitive information.

Authors’ response:

We acknowledge some repetition, but we think that Figure 1 (now Table 2) is useful because it groups the monogenic forms according to response to levodopa.

Point 6: Genes’ names must be in cursive.

Authors’ response:

We thank the reviewer for reminding us, we have gone through the manuscript ensured the genes names are italicized, it was originally italicized but this was removed during the formatting process by the journal.

Point 7: Some of the “Recently described Parkinson’s disease genes” are not so recent.

Authors’ response:

We thank the reviewer for the comment, all of the “recently described” genes have been discovered post 2015, with less information than well-established genes such as SNCA (1997), LRRK2 (2004), and VPS35 (2011), PRKN (1998), PINK1 (2004) etc. and so we think they should be grouped together, although we acknowledge this distinction is somewhat arbitrary.

Point 8: In X-linked Dystonia Parkinsonism reader doesn´t know what gene authors are talking about until TAF1 is named in pathophysiology.

Authors’ response:

Thank you for this comment. We have revised this paragraph to make this point more clear, including the following sentence:

“X-linked dystonia-parkinsonism (XDP), also referred to as Lubag, is a movement disorder initially described in Filipino males, caused by the insertion of a SINE-VNTR-Alu (SVA) type retrotransposon in intron 32 of the TAF1 gene [72,73].”

Point 9: Why authors talk about “Dual LRRK2 and GBA mutation carriers” so separately from the other genetic information?

Authors’ response:

We think this is a unique and interesting scenario of mutations in 2 different genes, and that it belongs nears the section on heterozygous carriers of autosomal recessive genes.

Point 10: In “Conclusion”, authors repeat the conclusions from other works.

Authors’ response:

Thank you for the comment, we have adjusted the Conclusion according to the other reviewers comments (see above), including a section on the therapeutic implications:

“It has become apparent that there are major ethnic and regional differences in the distribution of mutations in PD genes. There has been further evidence on the role of heterozygous carriers in autosomal recessive PD genes, and the effect of having mutations in both LRRK2 and GBA. Additionally, there is a suggestion that the underlying monogenic cause may influence the disease course as well as the response to levodopa and DBS.

Advances in genomic technology provide individuals with PD with greater access to genetic testing through both clinical and research pathways. Global efforts will play a key role in exploiting this genomic data. Worldwide studies can pool many patients to identify rare genetic causes of PD and can also be used to attempt to replicate important genetic discoveries. Furthermore, they offer greater representation of underrepresented populations from different ethnic groups and geographic regions. There are several major global projects to identify new disease genes in PD including established initiatives such as International Parkinson Disease Genomics Consortium [149] and newer initiatives such as the Global Parkinson’s disease genetics program (GP2) [150].“

Point 11: “GBA variants” must be reviewed. It has been shown that variants in GBA affect the phenotype of PD, but not only pathogenic variants.

Authors’ response:

We appreciate the reviewers comment, we have added the following sentence to the paper, citing a paper by Stoker et al.:

“Furthermore, a recent study showed that both pathogenic (i.e. associated with Gaucher disease) and non-pathogenic (i.e. not associated with Gaucher) variants in GBA are common in PD with a more aggressive course in terms of dementia and motor progression [125]."